# Untargeted Metagenomic Investigation of the Airway Microbiome of Cystic Fibrosis Patients with Moderate-Severe Lung Disease

**DOI:** 10.3390/microorganisms8071003

**Published:** 2020-07-04

**Authors:** Giovanni Bacci, Giovanni Taccetti, Daniela Dolce, Federica Armanini, Nicola Segata, Francesca Di Cesare, Vincenzina Lucidi, Ersilia Fiscarelli, Patrizia Morelli, Rosaria Casciaro, Anna Negroni, Alessio Mengoni, Annamaria Bevivino

**Affiliations:** 1Department of Biology, University of Florence, Sesto Fiorentino, 50019 Florence, Italy; giovanni.bacci@unifi.it (G.B.); francesca.dicesare@stud.unifi.it (F.D.C.); alessio.mengoni@unifi.it (A.M.); 2Cystic Fibrosis Center, Anna Meyer Children’s University Hospital, Department of Pediatrics Medicine, 50139 Florence, Italy; giovanni.taccetti@meyer.it (G.T.); daniela.dolce@meyer.it (D.D.); 3Centre for Integrative Biology, University of Trento, 38122 Trento, Italy; federica.armanini@unitn.it (F.A.); nicola.segata@unitn.it (N.S.); 4Children’s Hospital and Research Institute Bambino Gesù, 00165 Rome, Italy; vincenzina.lucidi@opbg.net (V.L.); evita.fiscarelli@opbg.net (E.F.); 5Cystic Fibrosis Center, IRCCS G. Gaslini Institute, Department of Pediatrics, 16147 Genoa, Italy; PatriziaMorelli@gaslini.org (P.M.); rosariacasciaro@gaslini.org (R.C.); 6Department for Sustainability, Italian National Agency for New Technologies, Energy and Sustainable Economic Development, ENEA Casaccia Research Center, 00123 Rome, Italy; anna.negroni@enea.it

**Keywords:** cystic fibrosis, airway microbiome, metagenome composition, antibiotic resistance genes

## Abstract

Although the cystic fibrosis (CF) lung microbiota has been characterized in several studies, little is still known about the temporal changes occurring at the whole microbiome level using untargeted metagenomic analysis. The aim of this study was to investigate the taxonomic and functional temporal dynamics of the lower airway microbiome in a cohort of CF patients. Multiple sputum samples were collected over 15 months from 22 patients with advanced lung disease regularly attending three Italian CF Centers, given a total of 79 samples. DNA extracted from samples was subjected to shotgun metagenomic sequencing allowing both strain-level taxonomic profiling and assessment of the functional metagenomic repertoire. High inter-patient taxonomic heterogeneity was found with short-term compositional changes across clinical status. Each patient exhibited distinct sputum microbial communities at the taxonomic level, and strain-specific colonization of both traditional and atypical CF pathogens. A large core set of genes, including antibiotic resistance genes, were shared across patients despite observed differences in clinical status, and consistently detected in the lung microbiome of all subjects independently from known antibiotic exposure. In conclusion, an overall stability in the microbiome-associated genes was found despite taxonomic fluctuations of the communities.

## 1. Introduction

The respiratory microbial composition is particularly relevant for CF patients. In fact, bacterial lung infections reduce life expectancy in most individuals with cystic fibrosis (CF) [1]. Many analyses of CF microbiota have been done (see for instance [2,3]). Most of these studies used 16S rRNA gene sequencing, yielding the identities and relative abundances of the taxa present (i.e., the microbiota). However, 16S rRNA based analyses have strong limits in providing strain-level or functional (meaning based on functional genes) information, which are more appropriately gained by metagenomics analyses [4]. Studying the microbial genetic repertoire, e.g., antibiotic resistance and virulence-related genes, with respect to clinical status or treatment can identify mechanisms of microbial persistence and pathogenesis [3,5,6]. However, up to now, most of these studies were cross-sectional, analyzing samples collected at a single time point from individual patients. Longitudinal studies that analyze serial samples obtained from individual patients over time allow a better assessment of the impact of these potentially confounding variables (including patient’s age, sex, lung disease stage, and antibiotic use) in constructing tractable models of the relationship between the dynamics of the lung microbial community and the disease progression. Longitudinal metagenomic investigations on the complete CF microbiome (microbiota and metagenome) are few and on a limited number of patients [7,8,9,10] or focused on specific metabolic functions [11]. The availability of microbiome data from longitudinal studies would allow to gain information for constructing systems-biology based models of microbiome evolution inside the CF patients, of potential relevance for patient’s treatment and prognosis [3].

The aim of this was to increase the knowledge on the temporal dynamics of CF microbiome, to allow gaining information on the whole community changes (as either strains/taxa or microbial functions). We focused on patients with moderate-severe lung disease, chronically infected by *Pseudomonas aeruginosa*. In fact, in a previous work [12], chronic infection with *P. aeruginosa* has been found to be associated with dysbiosis in the lungs of patients with CF. The authors suggested that the dominance of one species remodels the lung microbiota and may promote increased severity of CF lung disease. Consequently, patients infected by *P. aeruginosa* could be a good model for investigating microbiome changes over time in a heavily impacted ecology of the lung microbiome. Moreover, a more detailed taxonomic and functional analysis could help elucidating the mechanisms leading to chronic infection with *P. aeruginosa* and the microbial factors that contribute to the global changes of their lung microbiome. In the present study, a shotgun metagenomic approach was used [13] to detect the entire sputum microbial genomic repertoire down to the strain level [14].

A cohort of 22 patients with moderate-severe lung disease, grouped according to homozygosity versus heterozygosity for ΔF508 (also known as F508del) in the CFTR gene and chronically infected with *P. aeruginosa*, was selected and followed over 15 months during which 8 patients underwent exacerbation events. We aimed to determine the composition of sputum microbiomes for these patients when longitudinally sampled during the periods of stability and exacerbation, defining the relationship between clinical status, sputum microbial metabolic gene repertoire, and the antibiotic-resistance (AR) gene composition of sputum bacterial community, providing a previously unknown, high-resolution view of CF sputum microbiome dynamics.

## 2. Materials and Methods

### 2.1. Ethics Statement

The study was approved by the Ethics Committees of Children’s Hospital and Research Institute Bambino Gesù (Rome, Italy), Cystic Fibrosis Center, Anna Meyer Children’s University Hospital (Florence, Italy) and G. Gaslini Institute (University of Genoa, Genoa, Italy) [Prot. N. 681 CM of 2 November 2012; Prot. N. 85 of 27 February 2014; Prot. N. FCC 2012 Partner 4-IGG of 18 September 2012]. All participants provided written informed consent before the enrollment in the study. All sputum specimens were produced voluntarily. All procedures were performed in agreement with the “Guidelines of the European Convention on Human Rights and Biomedicine for Research in Children” and the Ethics Committee of the three CF Centers involved. All measures were obtained and processed ensuring patient data protection and confidentiality. Informed written consent was obtained from all participants in the study. Parents gave their consent for minors.

### 2.2. Characteristics of Enrolled Patients

Twenty-two adolescents and adults with moderate-severe lung disease (15 females and seven males) were enrolled in the study between October 2014 and March 2015 at three Italian CF Center (Table 1). The study subjects were selected based on eligibility criteria that included all of the following: (i) a diagnosis of CF, i.e., a sweat test showing sweat Cl > 60 mmol/L and two known CFTR mutations causing the disease with pancreatic insufficiency (elastase < 5 μg/g/feces) [15], (ii) aged more than six years, i.e., between 11 and 55 years, (iii) chronically infected with *Pseudomonas aeruginosa* according to the Leeds criteria [16] and iv) decline in %FEV_1_ in the previous three years before enrollment by measuring the difference between the best %FEV_1_ registered within the previous year and the best %FEV_1_ registered two-years before specimen collection, following the criteria previously reported [17]. Clinical status at the time of collection was designated as *baseline* (BL), when clinically stable and at their clinical and physiological baseline, *on treatment* (TR), at exacerbation-associated (additional) antibiotic treatments, and *at recovery* (RC), upon completion of (the additional) antibiotic treatment [18]. Forced expiratory volume in 1 s as a percentage of predicted (%FEV_1_) [19] were measured according to the American Thoracic Society and European Respiratory Society standards [20]. CFTR genotype, sex, age, and antibiotic treatment for each patient were reported in (Table 1 and Appendix A). During serial sampling, data (antibiotic usage and spirometry) were collected.

### 2.3. Sample Collection, Processing, DNA Extraction and Sequencing

A total of 79 sputum samples were obtained by spontaneous expectoration at baseline, exacerbation-associated antibiotic treatments and recovery status. Samples were processed according to standard methods as previously described [5,21]. Respiratory pathogens were identified using the conventional techniques reported in the guidelines, as previously described [21,22]. The number of samples, microbiological status at sampling and samplings following exacerbation events are reported in Appendix A. Sputum samples were washed in 5 mL PBS and then centrifuged (3,800 g) for 15 min. Resulting pellets were resuspended in 5–10 mL DNAse buffer (10 mM Tris-HCl pH 7.5; 2.5 mM MgCl2; 0.5 mM CaCl2, pH 6.5) with 7.5 µL of DNAse I (2000 Units/mL) per 1 mL of sample (15U/mL final), incubated for 2 h at 37 °C, and washed twice by pelleting at 3,800 g for 15 min and resuspending in 10 mL SE buffer (75 mM NaCl, 25 mM EDTA, pH 7.5). Pellets were then resuspended in 0.5 mL lysis buffer (20 mM Tris-HCl pH 8.0; 2 mM EDTA pH 8.0; 1% (v/v) Triton; 20 mg/mL Lysozyme final concentration), incubated for 30 min at 37 °C before extracting DNA with the MoBio PowerSoil DNA Isolation kit as per manufacturer’s instructions. Libraries were prepared with the Nextera XT DNA Library Preparation kit (Illumina) and sequenced on the HiSeq2500 apparatus (Illumina). Raw sequence data reported in this study have been deposited in the NCBI “Sequence Read Archive” (SRA) under the project accession PRJNA516870.

### 2.4. Basic Sequence Analyses

Sequence quality was ensured by trimming reads using StreamingTrim 1.0 [23], with a quality cutoff of 20. Bowtie2 [24] was used to screen out human-derived sequences from metagenomic data with the latest version of the human genome available in the NCBI database (GRCh38) as reference. Sequences displaying a concordant alignment (mate pair that aligns with the expected relative mate orientation and with the expected range of distances between mates) against the human genomes were then removed from all subsequent analyses.

### 2.5. Taxonomic Classification of Metagenomic Contigs

Assembled contigs were taxonomically classified using BLAST. First, all genomes available for each species detected with MetaPhlAn2 were downloaded from NCBI and used to build a database for each sample. All genomes reporting an identity higher than 90% and a coverage higher than 80% were collected and used for taxonomic classification. Contigs reporting hits with genomes coming from a single species were assigned to that species whereas contigs reporting hits from multiple species were flagged as unknown.

### 2.6. Bioinformatic and Statistical Analyses

To test for differentially distributed pathways and taxa across exacerbation events and genotypes we used a moderated t-test as implemented in the limma package [25], version 3.34.9. Data obtained with MetaPhlAn2 (taxonomic composition) and HUMAnN2 (pathway composition) were fitted into limma’s model using subjects as blocking variable. Since both software quantify biological units using relative counts (HUMAnN2 uses “copies per million” and MetaPhlAn2 uses percentages) we transformed this data into logarithmic values using the formula: log_2_(x + 0.1), where x are the relative counts. Obtained *p*-values were corrected using the Benjamini-Hochberg correction method. A similar approach has been used for antibiotic genes detect along assembled contigs. Here the number of reads that mapped onto each gene was used to estimate differentially abundant genes. Since the number of reads for each sample was variable (the ratio of the largest library size to the smallest was more than 10-fold) we used limma’s voom method [26] to fit our model, as suggested by the author of limma.

Metabolic and regulatory patterns were estimated using HUMAnN2 [27] and considering only those pathways with a coverage value ≥ 80%, whereas the taxonomic microbial community composition was assessed using MetaPhlAn2 [28]. The CARD database [29] was used in combination with the Resistance Gene Identifier (RGI, version 4.0.3) to inspect the distribution of antibiotic resistance gene (AR genes). Strain characterization was performed using StrainPhlAn [30].

Statistical analyses were conducted in the R environment [31] (version 3.4.4) with the help of external packages [25,26,32]. The taxonomical and functional composition on lung microbiome was explored using permutational multivariate analysis of variance (PERMANOVA with 1000 permutations), ‘adonis2’ function of vegan package version 2.5-2; whereas differences in bacterial diversity (Shannon and inverse Simpson) were tested using analysis of covariance (ANCOVA), ‘aov’ function. The model fitted for both analyses was:
X ~ Status + Genotype + Subject + FEV_1_ + days
where, Status is the exacerbation event, Genotype is the CFTR genotype, Subject is the patient, FEV_1_ was the forced expiratory volume in 1 s, and days, was the number of days from the enrolment in the study. For the ANCOVA analyses. Tukey’s post hoc tests were performed to test for mean differences within each factor used to build the full model (excluding FEV_1_ value and days since they were not categorical variable). Ordination analyses were conducted on both taxa (From MetaPhlAn2) and pathways (from HUMAnN2) using the function ‘ordinate’ of the phyloseq package [32] (version 1.23.1) with principle coordinate decomposition method (PCoA) and the Bray-Curtis dissimilarity index. The same index was used to inspect the distribution of samples and compare beta diversity level in both taxonomic composition and pathways.

## 3. Results

### 3.1. Population and Sampling

Twenty-two patients with CF were enrolled for a total of 15 females and seven males. The patients were chosen from a larger cohort of patients with moderate-severe lung disease (30 < %FEV_1_ < 70) and chronically infected by Pseudomonas aeruginosa. During the study period, they were treated with maintenance antibiotics (aerosol) and only a subset (*n* = 8) received clinical intervention in form of supplementary antibiotics (oral or/and intravenous) for a pulmonary exacerbation (CFPE) (Table 1 and Appendix A). The bacterial microbiome was investigated on sputum samples obtained every 3–4 months from 22 individuals along a survey of 15 months. Within the 22 subjects enrolled, two were lost to follow up, 8 underwent episodes of exacerbations, which provided the opportunity to explore the microbiome composition along the events. In total, 79 samples from these 22 subjects were collected and analyzed by a whole metagenomic sequencing approach.

### 3.2. Airway Microbiomes are Taxonomically Distinct and Show Patient-Specific Strain Colonization

The overall taxonomic representation of the microbiomes from the 79 samples is reported in Figure 1a,b, whereas a summary of obtained reads per sample was reported in Appendix A. Firmicutes, Proteobacteria, Bacteroidetes, and Actinobacteria were the most represented phyla. A high relative abundance (49% of total reads) of the “classical” CF bacterial signatures (taxa), such as *Staphylococcus aureus* and *Pseudomonas aeruginosa*, *Rothia mucilaginosa*, and *Prevotella melaninogenica* (all present in the top-10 species within each phylum, Figure 1b and Appendix A), was found.

Although principal coordinates analysis showed that subjects did not cluster based on treatment events and/or genotype (Appendix A), the PERMANOVA analysis (Table 2) reported a significant effect of both factors. However, the *R*^2^ values, namely the proportion of variance explained by the factor considered, were very low (Table 2, *R*^2^ = 0.03 for both factors, *p*-values < 0.05) probably due to intra-patient heterogeneity. No interaction effect of CFTR genotype on sputum microbiome was found (*p*-value > 0.05, treatment-genotype interaction effect). Subject effect was predominant with an R^2^ value of 0.52. Similarly, neither FEV_1_ nor time showed any significant relationship with taxonomy or functional profile (Table 2). A high fraction (more than 50%) of the total variance can be thus explained by inter-subject variation.

We then performed a metagenomic data-based strain-level analysis of the sputum microbiomes, by StrainPhlAn, a tool that permits to identify the specific strain of a given species within a metagenome [30]. This analysis demonstrated, in samples from the same patient but at different time points, that bacterial lineages were in general, closely related and tightly clustered together, confirming a patient-specific bacterial colonization and colonizing strain stability over time (Figure 2 and Appendix A).

Bacterial diversity measures (Shannon and inverse Simpson indices) varied according to clinical status, genotype, and subject (Appendix A). Samples collected during clinical treatments exhibited lower microbial diversity than samples collected at either baseline or recovery visits, highlighting the role of clinical treatments in perturbing CF lung communities as confirmed by the Tukey’s post hoc test.

### 3.3. Stability and Subject-Specific Distribution Patterns of Metagenomic Functions

The results of functional metagenomic analyses were consistent with the taxonomic findings described above. The list of metabolic pathways identified in metagenomic assemblies is reported in Appendix A. The category “Biosynthetic pathways” was the most represented functional category (Figure 3).

Pathways were mainly detected in members of the phyla Firmicutes and Proteobacteria, followed by Bacteroidetes and Actinobacteria. Exacerbation events and patient genotype significantly impacted pathway distribution (Table 2, *R*^2^ values of 0.04 and 0.03 respectively, *p*-values < 0.05), though with less an effect than that of subject (*R*^2^ = 0.48). The sample distribution according to representation and abundance of metabolic pathways was very heterogeneous with no sharp differences according to genotypes or exacerbation events (Appendix A). Alpha diversity of metabolic pathways dropped significantly in samples collected during exacerbation events, but the drop was significant only considering the inverse Simpson index (*p*-value = 0.036, Appendix A). Comparing beta-diversity values on both taxonomic and functional distribution a lower taxonomic similarity than functional (pathways) was detected (Figure 4 and Appendix A).

In other words, metabolic pathways were very consistent across patient status (baseline, treatment recovery) and had even less fluctuations than microbiota composition. This evidence was additionally confirmed by the differential abundance analysis. For contrasts made within each genotype, 40 pathways reported significant differences across exacerbation statuses [*p*-values < 0.05 and |log(fold-change)| > 5] all in the homozygote group (Appendix A), whereas, considering all samples together, no pathway was found to be more abundant in one condition in respect to another (data not shown). These results confirmed the resilience of the CF microbiome, suggesting that neither clinical change nor antibiotic treatments are accompanied by major changes in sputum microbial functions.

### 3.4. Resistome Composition through Exacerbation Events and Treatments

Antibiotic resistance genes (ARG) were inspected in relation to treatment events. Mapping of sequence reads to the CARD database resulted allowed to identify resistance-associated genetic determinants representing a range of resistance mechanisms, including antibiotic inactivating enzymes and efflux pumps, and conferring resistance to a number of antibiotic classes, including peptides, aminoglycosides, fluoroquinolones, monobactams, and nitroimidazoles.

Only six types of genes conferring resistance to β-lactamases and multidrug efflux transporters, were found to be affected by an exacerbation condition, all regarding samples from patients heterozygous for ΔF508 whereas, as found for metabolic pathways, no gene was significantly impacted in terms of abundance by antibiotic treatment when considering all samples at once (Appendix A). A similar approach was used to inspect the effect of antibiotic treatment on ARG distribution. ARG were inspected also in relation to the antibiotic treatments reported in Appendix A. The class of each antibiotic was correlated to the presence (and the abundance) of genes that may, in principle, confer resistance to antibiotics from the corresponding class. Differential abundance analyses (Table 3 and Appendix A) showed 11 genes affected by antibiotic intake (8 as reduction in abundance, 3 as increase in abundance), suggesting a corresponding decrease in the number of sensitive or increase in abundance of resistant strains. Results showed a large group of ARGs present in most of the samples and that the antibiotic treatment used in each sample was mirrored by the representation of the ARG classes (Figure 5 and Appendix A).

## 4. Discussion

Longitudinal studies provide important information on the stability and dynamics of microbial ecosystems [33] As all biotic communities, microbial communities tend to evolve towards a stable composition, either in natural environment or in association with host (as human-associated microbiomes). Changes in the community can be triggered by external conditions, as changes in host physiology (e.g., inflammation status) and/or other perturbations (e.g., antibiotic treatment). Indeed, perturbation studies help to probe community dynamics and resilience and possibly discover new findings for accessing ways for modifying the microbiome [34,35]. Here, we investigated the temporal dynamics of the CF sputum microbiome using shotgun metagenomics, including both periods of stability and respiratory exacerbations. Key questions were (i) what was the composition and stability of the lung microbiome in patients with CF when longitudinally sampled; and (ii) if the clinical status influenced the metabolic repertoire and resistome composition of lung bacterial community. Our results describe a unique examination of the dynamic of the lung microbiome in patients with moderate-severe lung disease carrying the ΔF_508_ mutation of CFTR gene and containing clinical measurements over a 15-month period.

The sputum microbiomes of CF patients were highly patient-specific, suggesting the host has one of the most important determinants of sputum microbiome composition. Indeed, there was less variation within the same individual at different time points than between different individuals at the same time point, proving some degree of temporal stability of an individual’s sputum microbiome, as indicated by the lack of a time effect on the taxonomic distribution of microbiomes. Moreover, the use of strain-level profiling allowed to monitor the resilience of predominant taxa detected in sputa of CF single patients during the entire study period. Assembly-free strain-level profiling in metagenomes through single nucleotide variants (SNVs) and genomic content has been widely used for comprehensive strain-resolved metagenomics [36]. Data derived from StrainPhlAn, a tool developed for the analysis of human microbiome that permit identification of the specific strain of a given species within a metagenome [30], have been found to correlate with traditional typing methods like MetaMLST, a metagenomic cultivation-free extension of Multi Locus Sequence Typing (MLST) [37].

Our results revealed that the predominant taxa detected in sputa of CF patients exhibited extraordinary resilience, as demonstrated by the presence of the same strains of several species during the entire study period. Resilience at the genus or species level was already known [38,39,40]. However, our approach went further, indicating that even single strains, not just more general taxonomies (as genus or species levels) are stable inside patients. Carmody and colleagues showed a relatively stable sputum community that was often altered during period of exacerbation even in the absence of viral infection or antibiotics only in a small group of patients [41]. A similar result was shown in the work from Fodor and colleagues [40] where, though occasional short-term compositional changes in the airway microbiota were found, the main taxonomic signatures of CF disease were highly stable. Even in other pulmonary diseases, such as non-cystic fibrosis bronchiectasis, respiratory sample bacterial communities showed a conserved structure for long periods of time, as showed in the work by Cox and colleagues where patients were followed for a six-month period [39].

Antibiotic exposure did not result in durable, persistent changes in sputum microbiota; the main taxa linked to CF infection were still present even after aggressive antibiotic treatment. From a taxonomic perspective, samples coming from the same patient clustered together, highlighting the role of the host in bacterial strain selection during the baseline but even during (and after) exacerbation events. Despite this patient-specific colonization, sputum taxonomic composition differed significantly from one subject to another even when sampled at the same time. Though the common CF pathogens were recovered, a notable exception was found for *Rothia mucilaginosa*. In fact, in contrast with other studies where this species was rarely identified [42,43,44], in our samples, it was detected in high relative abundance. This finding may suggest a potential involvement of *R. mucilaginosa* in CF microbiome dynamics and pathogenicity, which deserves further attention.

Conversely, microbial functional genetic pathways were more homogeneous across patients. This high conservation could be related to the characteristics of the lung environment itself, such as mucus compositions, nutrient availability, and oxygen levels, which can be broadly similar across patients with a similar clinical status. This finding is consistent with the concept that the function of a biotic community is more conserved than the presence of single members due to functional redundancy of different microbial taxa [45]. From this point of view, the airway microbiome can be considered as performing a similar “ecosystem service”, irrespective of the taxonomy present as pointed out by various authors in other environments [45]. Evidence of functional stability of the human microbiota was previously reported for the gut microbiome [46], indicating that single subjects can be considered to some extent as different ecological niches, inhabited by unique collections of microbial taxa (i.e., strains), but sharing the same set of genes. Investigations on the actual functionality (e.g., by metatranscriptomics) of the identified core-set of genes could provide clues about the genetic function of the microbiome to be targeted in future therapeutic treatments [3].

The finding that CFTR genotypes relate with different representation in some pathways, may suggest that the airways microbiome is influenced by the type of CFTR alteration. However, this hypothesis deserves further attention to clarify a putative role of microbial pathways with respect to the CFTR genotype and vice versa. If confirmed, this hypothesis could offer possible opportunities for treating patients by targeting some CFTR genotype-related microbial metabolism.

Despite a clear effect of antibiotic treatment during (and after) exacerbation periods, the community structure is always recovered with the main pathogenic taxa emerging again. This effect is confirmed by the resistome analysis of CF airway microbiota (i.e., all antibiotic-resistance genes in both pathogenic and non-pathogenic bacteria), by correlation of antibiotic-resistance genes (ARG) distribution and antibiotic intake. In the present study, patients subjected to a given antibiotic treatment did not seem to select bacteria resistant to the antibiotic used but the detection of a resistance genetic determinants seems to be distributed in almost all patients regardless of the treatment. Few studies have characterized the airway resistome associated with the airway microbiome in patients with CF lung disease by shotgun sequencing methods [5,47,48,49,50]. Understanding the impact of antibiotic treatment on the respiratory tract resistome could allow one to have a more in-depth comprehension of emergence and expansion of populations of multi-resistant organisms and spread of genes encoding antimicrobial resistance in CF airway microbiota [51].

## 5. Conclusions

In conclusion, the temporal dynamics of the sputum microbiome in the largest cohort of patients with CF analyzed so far, showed (i) patient-specific signatures of the airway microbiome at strain-level, (ii) lack of variation in the microbiome across pulmonary exacerbations, and (iii) a core set of antibiotic resistance genes that did not vary by antibiotic intake. While the dynamics of CF sputum microbial composition were highly patient-specific, the overall sputum metagenome composition was stable, showing a high resilience along time and antibiotic exposure. The high degree of redundancy in the CF lung microbiome could testify to ecological aspects connected to the disease that were never considered so far, as the large core-set of genes shared between patients despite observed differences in clinical status or antibiotic treatment. The main conclusion of the present study is that the management of chronic CF infection may be improved by a more patient-specific personalization of clinical care and treatment. In particular, moving away from taxonomic inventories towards gene content of CF microbiome could lead to the identification of the microbial gene repertoire associated with CF lung disease and may provide the clinicians with new biomarkers of CF progression and targets for antibiotic therapy [3,52]. Longitudinal studies of CF airway microbiota will permit to tailor therapeutic interventions and select antibiotic therapies based on the composition and relative abundance of antibiotic resistance genes within the respiratory microbiome. 

## Figures and Tables

**Figure 1 microorganisms-08-01003-f001:**
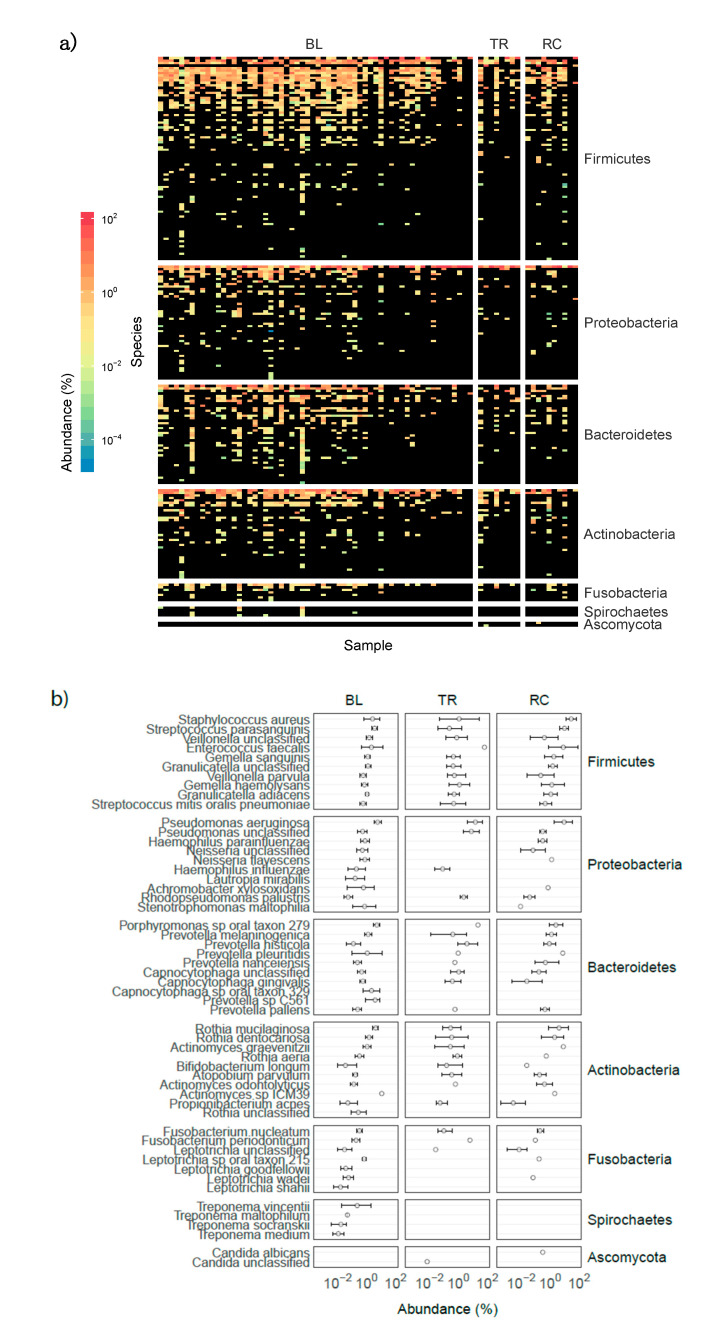
Taxonomic distribution in patients enrolled in the study. (**a**) The taxonomic distribution of all species detected using MetaPhlAn2 was reported in each row of the matrix whereas columns represent samples collected during the study. Colors from dark blue to red were used to report “copies per million” (CPM) values as obtained from HUMAnN2 with black reporting a CPM value of zero. The plot was divided according to patient status: BL, baseline; TR, treatment; RC, recovery. Species were ordered according to their mean abundance and grouped according to their Phylum. (**b**) The mean abundance value of the top-ten species (if available) detected within each Phylum was reported together with the standard error. The relative abundance of taxa is reported (Abundance %).

**Figure 2 microorganisms-08-01003-f002:**
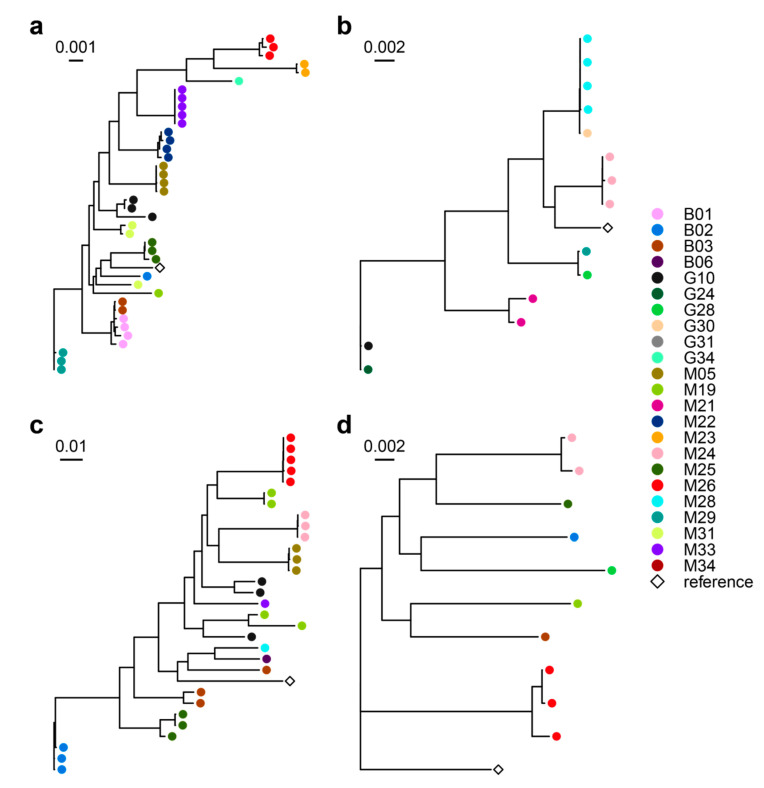
Strain-level phylogenetic trees of the main cystic fibrosis (CF) pathogens detected in the study. Phylogenetic trees obtained through StrainPhlAn pipeline were reported for the main pathogenic signatures of CF disease: (**a**) *Pseudomonas aeruginosa*; (**b**) *Staphylococcus aureus;* (**c**) *Rothia mucilaginosa*; (**d**) *Prevotella melaninogenica*. Points at the end of each clade are colored according to patients so that two points with the same color, in the same tree, represent the same species in two different time points, for the same patient.

**Figure 3 microorganisms-08-01003-f003:**
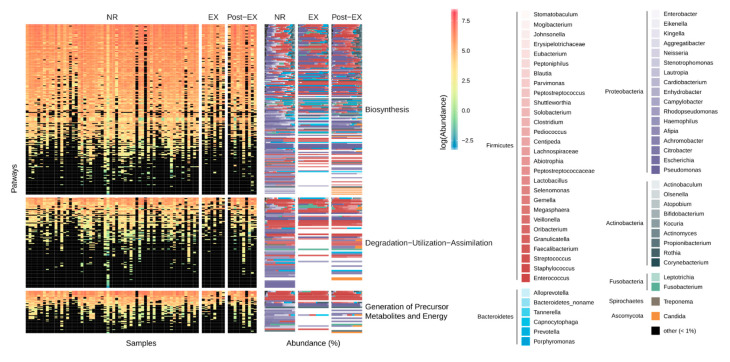
Pathway distribution according to exacerbation events. The pathway distribution was reported for each sample (columns) and for each pathway detected (rows). Colors from dark blue to red were used to report “copies per million” (CPM) values as obtained from HUMAnN2 with black reporting a CPM value of zero. On the right, the percentage of taxa in which each pathway was detected was reported using different colors. The main colors correspond to the Phylum whereas the different shades correspond to the genus detected (if available). BL, baseline; TR, treatment; RC, recovery.

**Figure 4 microorganisms-08-01003-f004:**
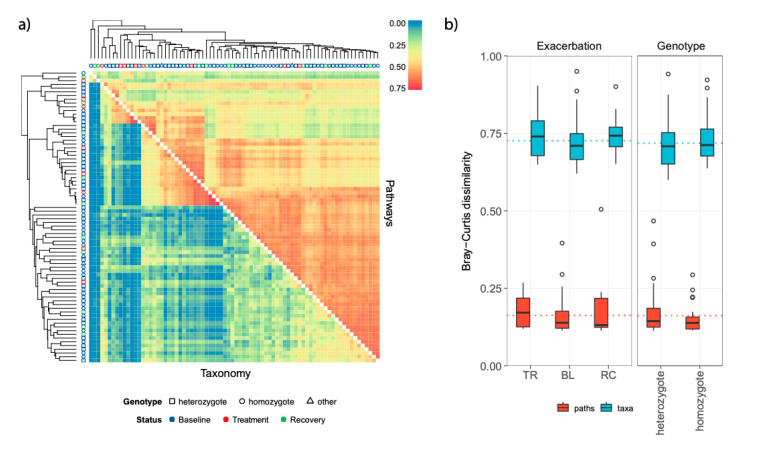
Beta diversity analysis on both taxonomic and functional distribution. (**a**) Hierarchical clustering based on UPGMA method. Clustering was performed on both pathway distribution (the upper triangle) and taxonomic composition of samples (lower triangle). The Bray-Curtis distance was used to compute distances between samples, but it was transformed into similarity value by subtracting 1 before plotting. Thus, red colors represent high similarity values whereas blue colors represent low similarity values. The shape of the points on each tip of trees refers to the genotype whereas the colors refer to the exacerbation events. (**b**) Results of Tukey’s post hoc test on beta diversity values across patient genotypes and exacerbation events. Contrasts were computed even to test differences between taxonomic distribution and pathways with taxa reporting higher level of beta diversity. Homozygote and heterozygote refer to ΔF508 mutation of CFTR gene. BL, baseline; TR, treatment; RC, recovery. Each box shows the “interquartile range” (IQR) that is the differences between the third and the first quartile of data (the 75th and the 25th percentile). Horizontal bars are medians whereas whiskers represent the minimum and maximum values defined as Q1 – (1.5 × IQR) and Q3 + (1.5 × IQR), respectively.

**Figure 5 microorganisms-08-01003-f005:**
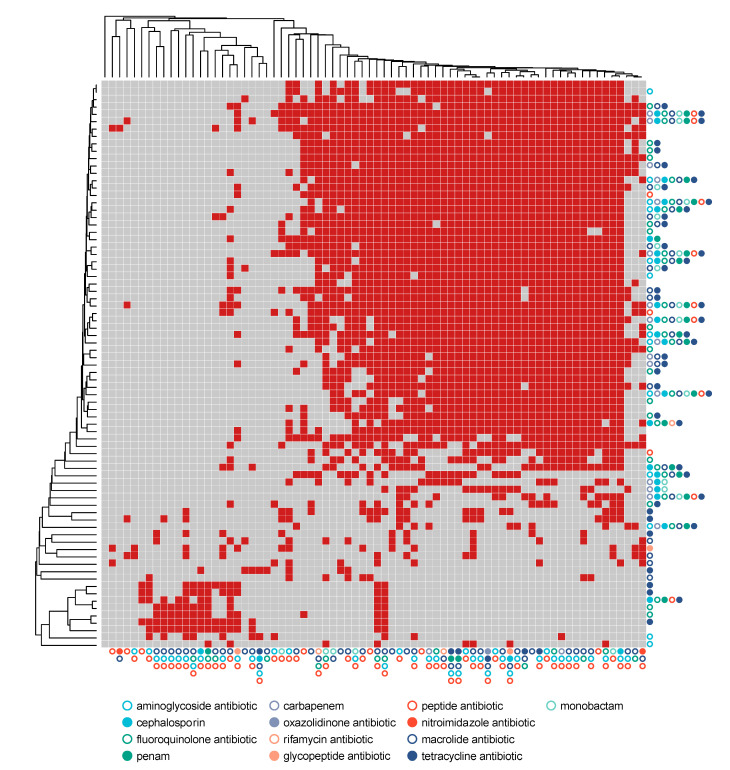
Antibiotic resistance genes map. Antibiotic Resistance Genes (ARGs) were reported in the y-axis whereas samples were reported in the x-axis. Antibiotic classes (both for ARG and for patient treatments) were reported using dots at the end of the heatmap. Hierarchical clustering was computed using the Jaccard index for binary data with the UPGMA method. Red cells correspond to the presence of a gene in samples whereas gray cell correspond to absence. Only genes detected in at least 10% of the subjects were reported.

**Table 1 microorganisms-08-01003-t001:** Characteristics of patients enrolled in the study.

ID	Genotype	Gender	FEV_1_ Status	Age	*n*	EX	%FEV_1_
B01	ΔF508/2183AA->G	M	S	27	5	yes	37.0 ± 1.70
B02	ΔF508/N1303K	F	SD	26	3	no	54.7 ± 3.48
B03	ΔF508/4016insT	F	S	30	4	no	55.0 ± 1.08
B06	ΔF508/ΔF508	F	SD	21	4	no	60.2 ± 3.42
G10	ΔF508/ΔF508	M	S	51	4	no	54.0 ± 3.08
G24	ΔF508/ΔF508	F	S	49	3	yes	31.0 ± 4.08
G28	ΔF508/ΔF508	F	NA	38	2	no	42.5 ± 1.50
G30	ΔF508/ΔF508	F	S	50	1	no	54
G31	G1244E/G42X	F	SD	53	2	no	41.5 ± 1.50
G34	ΔF508/ΔF508	F	S	39	1	no	47
M05	ΔF508/ΔF508	M	SD	32	4	no	34.8 ± 0.85
M19	ΔF508/ΔF508	M	S	24	4	no	44.0 ± 2.04
M21	ΔF508/N1303K	M	SD	27	4	yes	51.5 ± 4.35
M22	ΔF508/2789+5G->A	F	S	29	5	yes	50.4 ± 1.03
M23	ΔF508/G542X	F	S	30	4	yes	37.0 ± 1.47
M24	ΔF508/ΔF508	M	S	32	3	no	35.2 ± 0.85
M25	ΔF508/296+1G->T	F	SD	41	4	no	42.5 ± 2.02
M26	ΔF508/3849+10	F	SD	49	5	yes	39.6 ± 1.94
M28	ΔF508/N1303K	M	S	23	4	no	39.0 ± 1.08
M29	ΔF508/G542X	F	S	12	4	no	43.5 ± 3.75
M31	ΔF508/ΔF508	F	SD	11	3	yes	32.7 ± 4.41
M33	ΔF508/G85E	F	SD	13	5	yes	35.4 ± 5.78
**Total: 22**	**Heterozygote:11** **Homozygote:10** **Other:1**	**F:15** **M:7**	**S:12** **SD:9**	**32.1 ± 2.73**	**78**	**no:14** **yes:8**	**43.5 ± 1.09**

ID, study id; Genotype, cystic fibrosis transmembrane regulator (CFTR) genotype; Gender, gender; Age, enrollment’s age; n, number of samples collected; EX, yes if an exacerbation event has occurred during the study (no otherwise) [17] FEV_1_, mean value of forced expiratory volume in 1 s plus/minus the standard error on the mean; heterozygote and homozygote refers to ΔF508 genotype; %FEV_1_ status: S = with a rate decline lower than 1.5%, SD = with a rate decline higher than 5%. NA, not assigned.

**Table 2 microorganisms-08-01003-t002:** Permutational multivariate analysis of variance on both taxonomic distribution and metabolic pathways.

	Df	SumOf Sqs	*R* ^2^	F	Pr(>F)
TAXONOMY					
Status	**2**	**0.68**	**0.03**	**1.91**	**0.0300**
Genotype	**1**	**0.77**	**0.03**	**4.30**	**0.0020**
Subject	**18**	**11.97**	**0.52**	**3.74**	**0.0010**
FEV_1_ value	1	0.27	0.01	1.53	0.1349
Days	1	0.28	0.01	1.58	0.1229
Status:Genotype	1	0.11	0.01	0.64	0.7642
Residual	49	8.72	0.38	-	-
PATHWAY					
Status	**2**	**0.20**	**0.04**	**2.37**	**0.0220**
Genotype	**1**	**0.14**	**0.03**	**3.42**	**0.0080**
Subject	**18**	**2.43**	**0.48**	**3.20**	**0.0010**
FEV_1_ value	1	0.09	0.02	2.14	0.0989
Days	1	0.05	0.01	1.26	0.2458
Status:Genotype	1	0.08	0.02	1.96	0.1169
Residual	49	2.07	0.41	-	-

The permutational multivariate analysis of variance (PERMANOVA) analysis based on taxonomic distribution was reported in the upper part of the table whereas the analysis based on metabolic pathways was reported at the bottom. Df, degrees of freedom; SumOfSqs, sum of squares; *R*^2^, *r*-squared statistic (reported as proportion); F, F-statistic; Pr(>F), *p*-value associated to the F-statistic. Significant effects, namely those reporting a *p*-value lower than 0.05, were reported in bold.

**Table 3 microorganisms-08-01003-t003:** Antibiotic resistance genes differentially distributed depending on drug intake.

Gene Name	Gene Family	Resistance Mechanism	Drug Class	Antibiotic Class	logFC	AveExpr	t	*P.*Value	adj.*P*.Val
***basS***	pmr phosphoethanolamine transferase	antibiotic target alteration	peptide antibiotic	peptide antibiotic	−0.80	11.51	−5.22	<0.00001	0.0001
***FosA***	fosfomycin thiol transferase	antibiotic inactivation	Fosfomycin	peptide antibiotic	−1.10	10.08	−3.56	0.0006	0.0199
***ArmR***	resistance-nodulation-cell division (RND) antibiotic efflux pump	antibiotic efflux	aminocoumarin antibiotic; carbapenem; cephalosporin; cephamycin; diaminopyrimidine antibiotic; fluoroquinolone antibiotic; macrolide antibiotic; monobactam; penam; penem; peptide antibiotic; phenicol antibiotic; sulfonamide antibiotic; tetracycline antibiotic	peptide antibiotic	−1.56	9.24	−3.42	0.0010	0.0213
***OXA-50***	OXA beta-lactamase	antibiotic inactivation	cephalosporin; penam	aminoglycoside antibiotic	−0.45	11.61	−3.51	0.0008	0.0397
***Pseudomonas aeruginosa soxR***	ATP-binding cassette (ABC) antibiotic efflux pump; major facilitator superfamily (MFS) antibiotic efflux pump; resistance-nodulation-cell division (RND) antibiotic efflux pump	antibiotic efflux; antibiotic target alteration	acridine dye; cephalosporin; fluoroquinolone antibiotic; glycylcycline; penam; phenicol antibiotic; rifamycin antibiotic; tetracycline antibiotic; triclosan	fluoroquinolone antibiotic	−1.28	10.13	−4.43	<0.00001	0.0020
***MexR***	resistance-nodulation-cell division (RND) antibiotic efflux pump	antibiotic efflux; antibiotic target alteration	aminocoumarin antibiotic; carbapenem; cephalosporin; cephamycin; diaminopyrimidine antibiotic; fluoroquinolone antibiotic; macrolide antibiotic; monobactam; penam; penem; peptide antibiotic; phenicol antibiotic; sulfonamide antibiotic; tetracycline antibiotic	fluoroquinolone antibiotic	−0.73	11.11	−3.30	0.0015	0.0454
***MexR***	resistance-nodulation-cell division (RND) antibiotic efflux pump	antibiotic efflux; antibiotic target alteration	aminocoumarin antibiotic; carbapenem; cephalosporin; cephamycin; diaminopyrimidine antibiotic; fluoroquinolone antibiotic; macrolide antibiotic; monobactam; penam; penem; peptide antibiotic; phenicol antibiotic; sulfonamide antibiotic; tetracycline antibiotic	monobactam	1.75	10.90	4.95	<0.00001	0.0004
***mdtO***	major facilitator superfamily (MFS) antibiotic efflux pump	antibiotic efflux	acridine dye; nucleoside antibiotic	monobactam	2.13	10.40	3.73	0.0004	0.0150
***OpmD***	resistance-nodulation-cell division (RND) antibiotic efflux pump	antibiotic efflux	acridine dye; fluoroquinolone antibiotic; tetracycline antibiotic	monobactam	−2.15	10.44	−3.28	0.0016	0.0414
***MexT***	resistance-nodulation-cell division (RND) antibiotic efflux pump	antibiotic efflux	diaminopyrimidine antibiotic; fluoroquinolone antibiotic; phenicol antibiotic	monobactam	−1.52	10.57	−3.19	0.0021	0.0414
***MexK***	resistance-nodulation-cell division (RND) antibiotic efflux pump	antibiotic efflux	macrolide antibiotic; tetracycline antibiotic; triclosan	nitroimidazole antibiotic	−3.85	11.97	−3.71	0.0004	0.0407

Results of limma analysis were reported in the table together with the genotype, the antibiotic resistance gene name (according to the CARD database), and the antibiotic resistance class. Drug class refers to compounds with similar chemical structures and related mode of action which relies on the same resistance mechanism. Other columns are: logFC, the log2-transformed fold change value; AveExpr, the average log2-expression level for that pathway across all samples; t, the t-value according to limma’s moderated t-test; P.Value, the *p*-value; adj.P.Val, the adjusted p-value using the Benjamini–Hochberg correction. The fold-change value reported refers to the contrast between patients who are treated with an antibiotic sensible to the resistance mechanism specified and patients that were not. Only contrasts with an adjusted *p*-value lower than 0.05 and an absolute log fold-change value higher than 5 were reported.

## Data Availability

All data generated or analyzed during this study are included in this article. Raw sequence data reported in this study have been deposited in the NCBI “Sequence Read Archive” (SRA) under the project accession PRJNA516870.

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
