# Peer review of "Untargeted Metagenomic Investigation of the Airway Microbiome of Cystic Fibrosis Patients with Moderate-Severe Lung Disease"

_microorganisms, 2020, doi:10.3390/microorganisms8071003_

Round 1

Reviewer 1 Report

1.In this study the authors included 22 patients, which represent a very heterogeneous group, combined only by Pseudomonas infection. And the authors use methods of personalized medicine to analyze such a heterogeneous group. This leads to a number of observations that are unexpected for the authors. The authors suggest that Rothia mucilaginosa, and Prevotella melaninogenica are non-traditional CF taxa (Lines 124-126) For patients older than 30 years, Rothia and Prevotella are rare, but in children and adults under 30 in a stable state these are typical representatives of the lung microbial community.

2. The diagrams give an idea of the overall picture, but do not allow to understand which patient a particular sample belongs to.
For Figure 2, very similar colors were chosen to represent different patients.

3. The authors focus on antibiotic resistance genes, but the list itself is removed in the Supplementary. Since the CARD database is focused on nosocomial pathogens, most of the genes found belong to the P.aeruginose genome. At the same time, an analysis of other taxa contributing to the resistome is interesting.

4. A lot of interesting data of microbiome dynamics are in Supplementary and are not included in the Discussion

5. Lines 76-83 must be deleted because they contain technical information.

Author Response

REPLY TO REVIEWER 1

We are thankful to the Reviewer for the positive comments to improve the manuscript. We have considered his valuable suggestions and modified the manuscript, accordingly. In reply to your major comments, specific Answers are detailed below:

Reviewer’s comment:

1. In this study the authors included 22 patients, which represent a very heterogeneous group, combined only by Pseudomonas infection. And the authors use methods of personalized medicine to analyze such a heterogeneous group. This leads to a number of observations that are unexpected for the authors. The authors suggest that Rothia mucilaginosa, and Prevotella melaninogenica are non-traditional CF taxa (Lines 124-126) For patients older than 30 years, Rothia and Prevotella are rare, but in children and adults under 30 in a stable state these are typical representatives of the lung microbial community.

ANSWER: Done. We modified the Introduction accordingly, including Rothia and Prevotella in “classical” CF bacterial pathogens (see page 5, lines 201-203). “A high relative abundance (49% of total reads) of the “classical” CF bacterial signatures (taxa), such as Staphylococcus aureus and Pseudomonas aeruginosa, Rothia mucilaginosa, and Prevotella melaninogenica (all present in the top-10 species within each phylum, Fig. 1b and Table S3), was found”.

2. The diagrams give an idea of the overall picture, but do not allow to understand which patient a particular sample belongs to. For Figure 2, very similar colors were chosen to represent different patients.

ANSWER: We agree with the Reviewer. Color palette was modified, however since patients are 22, some colors may still be similar depending on screen color resolution.

3. The authors focus on antibiotic resistance genes, but the list itself is removed in the Supplementary. Since the CARD database is focused on nosocomial pathogens, most of the genes found belong to the P. aeruginose genome. At the same time, an analysis of other taxa contributing to the resistome is interesting.

ANSWER: We thank the Reviewer for this comment. Accordingly, we revised Results and Discussion sections (see pages 10 and 11, lines 299-306, lines 310-319; page 15, lines 410-419). Also, we moved into the main text the formerly Table S12 “Antibiotic resistance genes differentially distributed depending on drug intake” (now, Table 3).

4. A lot of interesting data of microbiome dynamics are in Supplementary and are not included in the Discussion

ANSWER: We agree with the Reviewer. Since in the new version of the manuscript part of the supplementary data were moved to the main text, we have now improved the Discussion.

5. Lines 76-83 must be deleted because they contain technical information.

ANSWER: Done

Reviewer 2 Report

Microorganisms_826830

Untargeted metagenomic investigation of the airway microbiome of cystic fibrosis patients with moderate-severe lung disease

This study examined the metagenomics of sputa from 22 CF patients over a time-period of 15 months. The longitudinal aspects of this study provided important information about the general stability of the microbiome in these patients, despite changes in their clinical status over time. It is a clearly-written study, with appropriate methodology and detailed experimental analysis. It provides further useful additional information to what is already known about CF microbiome.

My one major comment would be on the analysis of microbes at strain level. Was the data derived from StrainPhlAn correlated with traditional typing methods (for example pulsed-field gel electrophoresis/MLST/VNTR)? If so it would be worth mentioning this. If not, it may be worth adding a sentence highlighting how this this programme derives this information and how this might compare to traditional methods.

Minor comments:

Lines 76-83: These lines seem to have been inserted from the guide to authors and therefore need to be removed!

Line 107: Sputum samples

Line 120: 78 samples stated here, but in the abstract 79 samples are mentioned.

Fig 3-there is a problem with the wording after “degree” on this figure.

Line 187-upper, rather than upped?

Line 225-gap needed between “showed” and “11”.

Line 259-vice versa

Lines 274-275: A line from the guide to authors needs to be removed.

Author Response

REPLY TO REVIEWER 2

We thank the reviewer for the constructive comments and suggestions. We have carefully considered them and changes have been done accordingly. In reply to your major and minor comments, specific Answers are reported below.

Reviewer’s comment:

This study examined the metagenomics of sputa from 22 CF patients over a time-period of 15 months. The longitudinal aspects of this study provided important information about the general stability of the microbiome in these patients, despite changes in their clinical status over time. It is a clearly-written study, with appropriate methodology and detailed experimental analysis. It provides further useful additional information to what is already known about CF microbiome.

My one major comment would be on the analysis of microbes at strain level. Was the data derived from StrainPhlAn correlated with traditional typing methods (for example pulsed-field gel electrophoresis/MLST/VNTR)? If so it would be worth mentioning this. If not, it may be worth adding a sentence highlighting how this this programme derives this information and how this might compare to traditional methods.

ANSWER: We thank the Reviewer for this comment. We specified how the programme derives its finding and the reliability of the obtained data with respect to a MLST. Accordingly, we improved the discussion by adding the following sentences (page 14, lines 360-366): “Assembly-free strain-level profiling in metagenomes through single nucleotide variants (SNVs) and genomic content has been widely used for comprehensive strain-resolved metagenomics (Segata N. On the Road to Strain-Resolved Comparative mSystems. 2018;3(2):e00190-17). Data derived from StrainPhlAn, a tool developed for the analysis of human microbiome that permit to identify the specific strain of a given species within a metagenome, have been found to correlate with traditional typing methods like MetaMLST, a metagenomic cultivation-free extension of Multi Locus Sequence Typing (MLST) (Zolfo et al. Profiling microbial strains in urban environments using metagenomic sequencing data. Biol Direct 13, 9 (2018). More details are reported at the link http://segatalab.cibio.unitn.it/tools/strainphlan/

Minor comments:

Lines 76-83: These lines seem to have been inserted from the guide to authors and therefore need to be removed!

ANSWER: Done

Line 107: Sputum samples

ANSWER: Done

Line 120: 78 samples stated here, but in the abstract 79 samples are mentioned.

ANSWER: Done

Fig 3-there is a problem with the wording after “degree” on this figure.

ANSWER: Done. There was a problem in format conversion from the original Figure

 Line 187-upper, rather than upped?

ANSWER: Done

Line 225-gap needed between “showed” and “11”.

ANSWER: Done

Line 259-vice versa

ANSWER: Done

Lines 274-275: A line from the guide to authors needs to be removed.

ANSWER: Done

Reviewer 3 Report

Summary: Here the authors aim to present longitudinal microbiome data collected from 22 patients with moderate-severe lung disease colonized with Pseudomonas Aeruginosa over a period of up to 15 months. Using a shotgun metagenomic approach, they aimed to describe observed temporal changes in CF sputum from each of the individual patients. This study is of interest based on the number of patients included here for this type of analysis.

Major Issues:

  • General Editing Issues: This should have been proofread before submission. In addition to addressing English language and flow for this narrative, the authors left notes with instructions on manuscript preparation that need to be removed. (For example, Section 1. Introduction. Lines 76-83 and Section 5. Lines 274-275).
  • Presentation of Methods and Data: This needs to be strengthened overall. Special care should be used to ensure patient/participant confidentiality. The listing of the specific hospital for each patient, in addition to sex and age I think provides more information than needed and can lend itself to possible participant identification. Additionally, the further clarification of the sputum collection time points – is it not helpful to include patients with only one time point if the novelty of the study lies in the longitudinal multi-point follow up. 
  • I also wonder if this manuscript format/length is appropriate for the number of results shared. The paper is relatively short and I think that the shorter length has an impact here on the quality of methods/results/discussion.
  • Patient characteristics need be more fully described in the text and not just left to supplementary material. It is recognized that age, sex, and use of antibiotics significantly impacts the microbiome and this information should be included in the presentation and in the discussion.
  • It is well documented that the pathogenic landscape varies with age particularly in CF. The variation of age in these subjects (11-51) needs to be addressed specifically. Additionally, the consent process especially when dealing with minors should be described clearly in the text.
  • CFTR Genotype is important information. The authors discuss the impact of CFTR genotype on microbial pathways in the discussion, but do not focus on this data in the remainder of the text. If not in table form, summary statements should be made in the patient characteristics. Or perhaps make this into another separate manuscript where you can focus on this information more ? Also of important note related to genotype would be to address whether patients were on modulator therapy, did this change if they were started on modulator therapy during the course. What was the time period during which the clinical samples were collected?

Author Response

REPLY TO REVIEWER 3

Thank you very much for your careful reading and useful comments. Based on your comments, we have carefully revised our manuscript, adding some context, re-writing some parts and expanding some explanations. We hope we have addressed your concerns. Detailed reply to your comments:

Reviewer’s comment:

Summary: Here the authors aim to present longitudinal microbiome data collected from 22 patients with moderate-severe lung disease colonized with Pseudomonas Aeruginosa over a period of up to 15 months. Using a shotgun metagenomic approach, they aimed to describe observed temporal changes in CF sputum from each of the individual patients. This study is of interest based on the number of patients included here for this type of analysis.

Major Issues:

General Editing Issues: This should have been proofread before submission. In addition to addressing English language and flow for this narrative, the authors left notes with instructions on manuscript preparation that need to be removed. (For example, Section 1. Introduction. Lines 76-83 and Section 5. Lines 274-275).

ANSWER: We thank the Reviewer for this comment. The presence of “notes with instructions on manuscript preparation”, as those reported in the word template, was due to the automatic conversion of the word file to the template of Microorganisms Journal. We have proofread the manuscript again before submission. English language was checked and improved where necessary.

Presentation of Methods and Data: This needs to be strengthened overall. Special care should be used to ensure patient/participant confidentiality. The listing of the specific hospital for each patient, in addition to sex and age I think provides more information than needed and can lend itself to possible participant identification. Additionally, the further clarification of the sputum collection time points – is it not helpful to include patients with only one time point if the novelty of the study lies in the longitudinal multi-point follow up.

ANSWER: We thank the Reviewer for this comment. We performed the required amendments by including the previous Supplementary Methods (Patients and sampling, and Inclusion criteria) into the main text. Also, the former Table S1 has been moved to the main text as Table 1. We removed the column “Hospital” from Table 1 to avoid possible participant identification.

I also wonder if this manuscript format/length is appropriate for the number of results shared. The paper is relatively short and I think that the shorter length has an impact here on the quality of methods/results/discussion.

ANSWER: We are glad that the Reviewer appreciated our work. We improved the manuscript in the Material and Methods, Results and Discussion section. In particular, we moved both results and associated methods from the Supplementary material to the main text (see pages 2-5), improving the Result section by adding a new paragraph (3.1 Population and sampling), including Table 2 (formerly Table S5) and giving a more in-depth description of resistome composition in a new paragraph (3.4 Resistome composition through exacerbation events and treatments, pages 11-12), including Table 3 (formerly Table S12). Discussion section was revised and improved to take into account the results now shown in the main text (see pages 13-15).

Patient characteristics need be more fully described in the text and not just left to supplementary material. It is recognized that age, sex, and use of antibiotics significantly impacts the microbiome and this information should be included in the presentation and in the discussion.

ANSWER: We agree with the Reviewer. We included a new paragraph in the Results section (see 3.1 “Population and sampling”), and moved into the main text the former Table S5 (new Table 2).

It is well documented that the pathogenic landscape varies with age particularly in CF. The variation of age in these subjects (11-51) needs to be addressed specifically. Additionally, the consent process especially when dealing with minors should be described clearly in the text.

 ANSWER: We agree with the Reviewer. However, tough important, the number of patients here studied (22) cannot allow to stratify them into sound age categories, to decipher if age can have an impact on microbiome dynamics. The suggestion of the referee warrants a newly designed study with extensive sampling of patients stratified in relatively few age categories.

Two sentences related to the informed consent have been added into the Materials and Methods (Ethics statement, page 2, lines 86-87).

CFTR Genotype is important information. The authors discuss the impact of CFTR genotype on microbial pathways in the discussion, but do not focus on this data in the remainder of the text. If not in table form, summary statements should be made in the patient characteristics. Or perhaps make this into another separate manuscript where you can focus on this information more?

ANSWER: We thank the Reviewer for the specific comment. Results related to association with genotype are presented in Table 1, Table 2 and Figure 4. More specifically, Figure 4 in the main text summarizes the effect of genotype (Homozygote and heterozygote refer to DF508 mutation of CFTR gene) on the microbial pathways. Also, the PERMANOVA analysis confirmed the effect of genotypes in shaping the pathway distribution of CF lung microbiome with (Table 2) (R2 values = 0.03). The genetic characteristics of enrolled patients are reported in the new Table 1 (formerly Table S1). We focused on the present manuscript the obtained results and no further manuscript is in preparation.

Also of important note related to genotype would be to address whether patients were on modulator therapy, did this change if they were started on modulator therapy during the course. What was the time period during which the clinical samples were collected?

 ANSWER: Thanks for the comment. It is well known that CFTR modulators can impact the CF airway microbiome (Rogerts et al. J Cyst Fibros. 2019;S1569-1993), having CFTR-dependent and CFTR-independent effects on the microbiome. All enrolled patients were not receiving modulator therapy, neither before the enrollment nor during the follow-up. Patients were enrolled between October 2014 and March 2015 and samples were collected over 15 months of follow-up.